# Understanding the Unique Role of Phospholipids in the Lubrication of Natural Joints: An Interfacial Tension Study

**Aneta D. Petelska** [1,*]**, Katarzyna Kazimierska-Drobny** [2]**, Katarzyna Janicka** [1]**, Tomasz Majewski** [3] **and Wiesław Urbaniak** [2,*]

[1] Institute of Chemistry, University of Bialystok, Ciolkowskiego 1K, 15-245 Bialystok, Poland; k.janicka@uwb.edu.pl

[2] Faculty of Mathematics, Physics and Technical Sciences, Kazimierz Wielki University, J.K. Chodkiewicza 30, 85-867 Bydgoszcz, Poland; kkd@ukw.edu.pl

[3] Institute of Armament Technology, Faculty of Mechatronics and Aerospace, Military University of Technology, W. Urbanowicza 2, 00-908 Warsaw, Poland; tomaszmajewski@wat.edu.pl

[*] Correspondence: aneta@uwb.edu.pl (A.D.P.); wurban@ukw.edu.pl (W.U.); Tel.: +48-85-73-88-261 (A.D.P.); +48-52-32-57-641 (W.U.)

**Abstract:** Some solid lubricants are characterized by a layered structure with weak (van der Waals) inter-interlayer forces which allow for easy, low-strength shearing. Solid lubricants in natural lubrication are characterized by phospholipid bilayers in the articular joints and phospholipid lamellar phases in synovial fluid. The influence of the acid–base properties of the phospholipid bilayer on the wettability and properties of the surface have been explained by studying the interfacial tension of spherical lipid bilayers based on a model membrane. In this paper, we show that the phospholipid multi-bilayer can act as an effective solid lubricant in every aspect, ranging from a 'corrosion inhibitor' in the stomach to a load-bearing lubricant in bovine joints. We present evidence of the outstanding performance of phospholipids and argue that this is due to their chemical inertness and hydrophilic–hydrophobic structure, which makes them amphoteric and provides them with the ability to form lamellar structures that can facilitate functional sliding. Moreover, the friction coefficient can significantly change for a given phospholipid bilayer so it leads to a lamellar-repulsive mechanism under highly charged conditions. After this, it is quickly transformed to result in stable low-friction conditions.

**Keywords:** amphoteric articular cartilage; friction coefficient; interfacial tension; deactivation of surface-active phospholipid; $\beta_2$-Glycoprotein 1 ($\beta_2$-GP-1)

## 1. Introduction

In the study of aqueous solutions and phospholipid chemistry, it has been established that molecular self-assembly is the key to their functionality as the noncovalent interactions of hydrophobic forces facilitate the assemblage of molecules. The typical examples of these physicochemical representatives with functional surfaces include surfactant molecules that are assembled as liposomes, bilayers, and membranes. The surfaces formed by these low-friction lubricating agents in the body system are comprised of an interface and colloid aggregates. The surface physics are determined by the interrelationships between interfacial tension, wettability, and friction coefficient. On the other hand, the surface chemistry (e.g., that of the surface of articular cartilage) depends on the rate of adsorption, hydration, liposomes, and bilayer formation as well as the nature of the lamellar phospholipid phases.

Ultimately, the behavior of a solution-based interface is affected by the surface charge, dipoles and their distribution within the electric double layers, which are formed by the present structural molecular species. Numerous types of lamellar bodies/structures built by the molecular species have been identified in different cells and tissues and are classified according to their specific tissue or cellular function, including (a) hydrophilic surface coating and protectives (e.g., in lung alveoli), (b) hydrophobic protective barriers (e.g., in skin), (c) hydrophobic protective linings (e.g., in the stomach), and (d) hydrophilic sliding (e.g., in the articulating joints). These multilamellar phospholipid membranes (a–d) are commonly classified as lamellar bodies [1]. According to Hills' 'biological lubrication model' [2] and our lamellar-repulsive mechanism [3], phospholipids constitute the main solid-phase component in the diarthrodial joint tribosystem.

Many scientists have been able to apply knowledge about natural surfaces in relative motion, and their friction or lubrication and wear, to the creation of new lubricants, such as surface-grafted polymer brushes [4–12]. Bayer [13] has presented information about new materials and modified lubricin structures for low friction. This review presents the latest advances in understanding the function of lubricin in joint lubrication and documents previous achievements in transforming this biomedical knowledge into a new polymer design for advanced engineering tribology.

In this paper, we study the physics and chemistry of the articular surface of natural bovine knee cartilage. These include the measured key parameters/characteristics, such as hydrophilic/hydrophobic and amphoteric characteristics, interfacial tension, charge density, and friction coefficients of the surface under varying pH and friction loading conditions in buffer solutions. We introduce the mechanism of degenerative joint diseases and the destruction of phospholipid bilayers. The surface cartilage deterioration with the deactivation of surface-active phospholipid bilayers is seen to activate β2-Glycoprotein 1 through interactions. The interaction of the β2-Glycoprotein 1 ($-NH_3^+$) group and the phospholipid ($-PO_4^-$) group of ($-NH_3^+$) + ($-PO_4^-$) → ($-NH_3^+\ PO_4^--$) is strong enough to deactivate the bilayer surface of the phospholipids.

*Hydrophilic and Hydrophobic Character of the Phospholipid Membrane*

In phospholipid membranes, the wetted surfaces (pH of 6.5) are negatively charged ($-PO_4^-$). The differences in the charge density of the phosphoric groups ($-PO_4^-$) allow us to determine the wettability of the natural surface. Therefore, the wettability of the hydrated surface is described by the changes in the concentration of negatively charged phosphate groups ($-PO_4^-$), which are deactivated when the surface is dehydrated. Under such conditions, the hydrophobic groups that form a hydrophobic monolayer are activated, which is shown in Figure 1.

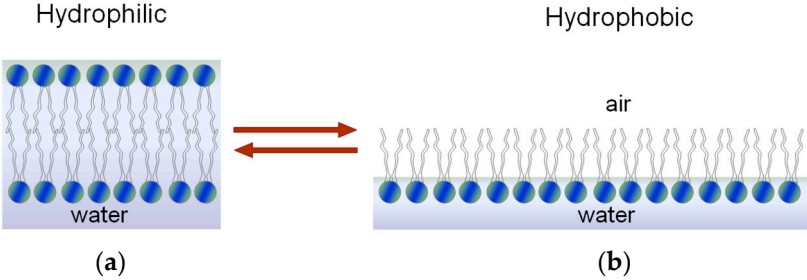

Hydrophilic　　　　　　　　　　　　　　　　　　Hydrophobic

air

water　　　　　　　　　　　　water

(**a**)　　　　　　　　　　　　　　　　　　　　(**b**)

**Figure 1.** The intelligent building of the surface of an articular cartilage phospholipid bilayer under (**a**) wet and (**b**) the air-dry conditions.

A change in the surface tension leads to conformational changes in the surface of phospholipids, with the surface changing from a bilayer (hydrophilic) to a monolayer (hydrophobic). The hydrophilic surface during dehydration (such as by air drying) has a slow increase in wettability, which indicates a conformational change of surface phospholipid molecules (flip-flop). An inadequate lubrication of joints in animals, particularly on the surface of the articular cartilage, can be attributed to the

deterioration of the bilayer area, wherein the wettability or the contact angle (θ) varies from 100° to less than 70° [2,14–16].

The morphological evidence demonstrating the phospholipid bilayers as the outermost lubricating lining of the articular surface is presented in Figure 2.

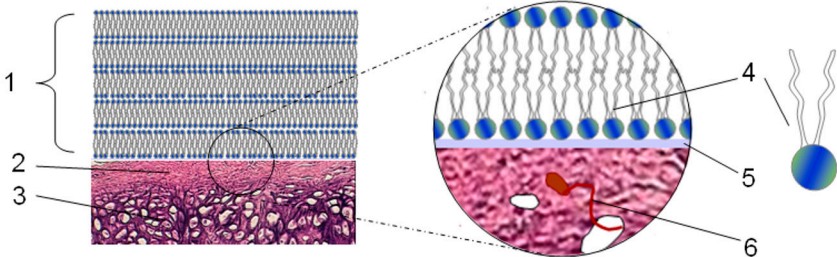

**Figure 2.** Images of the multilamellar lining of the adsorbed bilayers of a phospholipid of the articular surface of a human knee, where 1 indicates the phospholipid bilayers (from 2 to 5); 2, compacted bone tissue; 3, spongy bone tissue; 4, the phospholipid, 5, articular cartilage (AC); and 6, a blood vessel.

Using a high-resolution imaging technique, which was namely atomic force microscopy (AFM), a previous study established that a well-organized surface-active phospholipid covers the surface of the cartilage layer in a lamella-like arrangement as previously described by Hills [2,3,16] (Figure 3a). The wiping of the articular surface with lipid rinsing reagent results in a drastic removal of the surface amorphous multilayer [3] (Figure 3b). In mammals, the intact lipid layer of cartilage is lost during degeneration, thus affecting the efficient lubrication of the joint.

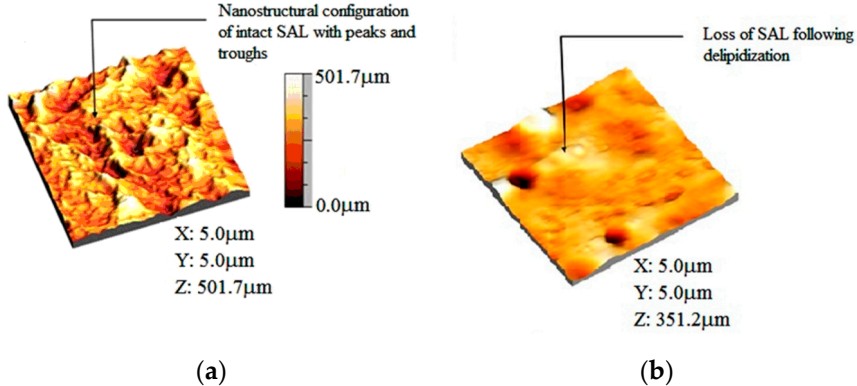

(**a**)        (**b**)

**Figure 3.** Topographic 3D image of healthy cartilage and lipid-free cartilage: (**a**) normal healthy cartilage after image processing, showing a nanostructured amorphous surface layer with several peaks and depressions; and (**b**) lipid-free cartilage after image processing, showing membrane loss overlay (surface layer at isoelectric point) of the articular surface (length (*X*) and width (*Y*) of the scanned area and the average height of the peak of the amorphous multi-layer surface (SAL)).

In this paper, we present data that sheds more light on the surface physics and chemistry of the articular surface of natural bovine knee cartilage. This involves measuring key parameters/characteristics, such as interfacial tension, charge density, and the friction coefficient of the surface, under varying pH and friction loading conditions in a universal buffer solution.

## 2. Materials and Methods

Phosphatidylcholine (PC, 99%) from Fluka (Neu-Ulm, Germany) with a molecular formula of $C_{40}H_{80}NO_8P.H_2O$ was used in the experiment. The phosphatidylcholine was isolated from hen egg yolk (fatty acid composition: 16:0–33%, 18:0–4%, 18:1–30%, 18:2–14%, and 20:4–4%).

The forming solution contained 20 mg/cm$^3$ phosphatidylcholine in n-decane:butanol (3:1). First, phosphatidylcholine was dissolved in chloroform to prevent oxidation. After this, the solvent was evaporated in an argon environment. The residue was dissolved in the n-decane:butanol solution (Polish Chemical Reagents), which was further purified by distillation. This resulted in $\varepsilon$ = 1.991 (293.15 K).

Buffers with pH values of 2–12 were prepared by Britton and Robinson [17] and were used as the electrolyte. The buffer was prepared by the addition of 0.2 M sodium hydroxide to 100 mL of solution with the following composition: 0.04 M 80% acetic acid, 0.04 M phosphoric acid, and 0.04 M boric acid (all chemicals from Avantor Performance Materials Poland, Gliwice, Poland). Determining whether the buffer had a suitable pH was based on the amount of added sodium hydroxide. The initial pH of the prepared buffer was 1.81. After the addition of 20 cm$^3$ NaOH (Avantor Performance Materials Poland), the pH changed, i.e., to 6.80 after the addition of 50 cm$^3$. In the experiments, the Britton and Robinson buffer [17] was used, which is a buffer used in standard biochemical analysis due to its wide pH range (2–12) and its composition that does not affect the biological membrane. The pH of the electrolyte solution was controlled during all measurements [18].

## 2.1. The Interfacial Tension Experiments

The interfacial tension $\gamma$ of the PC bilayer was calculated using the measurement results of the curvature radius, $R$, when the surface was convex, and a pressure difference, $\Delta p$, was placed on both sides. The method based on the Young and Laplace's standard equation was applied but the molecular-level peculiarities were neglected, which had a negligible effect on the calculations of $\gamma$. Hence, the following formula was used [19]:

$$2\gamma = \Delta pR \tag{1}$$

The method of obtaining interfacial tension measurements and the apparatus used to perform the experiments have been previously described [20,21]. The lipid membranes were formed in a Teflon diaphragm with an inner diameter of 1.5 mm, which had a certain amount of electrolyte on both sides. They were created according to the Mueller and Rudin method [22]. Interfacial tension was measured on a freshly prepared phosphatidylcholine bilayer 12 to 15 times. From all the readings of the instrument, we calculated the arithmetic mean and standard deviation values. All measurements of the preparation of the electrolyte solution were performed 2–3 times to check the repeatability of the interfacial tension value determinations [18].

## 2.2. Friction Coefficient versus Charge Density of the Cartilage Surface

The articular cartilage samples were collected from bovine knees that were about 1.5 years old. Osteochondral plugs with diameters of 5 and 10 mm were obtained from the lateral and medial condyles of the femurs. The cartilage discs were cut into three-millimeter earplugs with full adhesion to the underlying bone. The samples were stored at 253 K in 0.155 M NaCl (pH = 6.9) and fully thawed before testing.

After this, the discs were glued to the surface of stainless-steel plates and pins before friction tests were carried out in a universal buffer solution. Friction loading of the cartilage samples in different solutions with a pH of 2.0–9.5 was carried out using a Britton–Robinson [8] universal buffer solution. This consisted of a mixture of 0.04 M $H_3BO_3$, 0.04 M $H_3PO_4$, and 0.04 M $CH_3COOH$, and was adjusted to the desired pH with 0.2 M NaOH. A radiometer pH-meter with an electrode (Schott-BlueLine 16 pH type) was used in the experiment. This instrument was calibrated according to the recommendations of the International Union of Pure and Applied Chemistry (IUPAC). A detailed description of the friction coefficient measurements has been presented in a previous paper [23].

*2.3. Friction Test in Universal Buffer Solutions (pH 2.0–9.5)*

Measurements were carried out using a sliding pin-on-disc tribotester T-11 manufactured by NISTR, Radom, Poland. Tests were carried out at room temperature at a speed of 1 mm/s for 5 min with a load of 15 N (1.2 MPa), which corresponds to physiological lubrication [15].

The measurements of the cartilage/cartilage tribopair friction coefficients were obtained in a pH range of 2.5–9.5. Samples were equilibrated with each buffer solution under a certain load for 5 min and the friction coefficient was calculated in the pH function. Five tests were conducted using fresh samples for each experimental set-up with at least four repetitions per specimen pair, from which the mean and standard deviation were calculated.

## 3. Results and Discussion

*3.1. The Amphoteric Character of the Phospholipid Bilayer Surface of Spherical Lipid Bilayers Formed from Phosphatidylcholine*

The composition of lipids in the cartilage surface layer included 41% phosphatidylcholine, 27% phosphatidylethanolamine, and 32% sphingomyelin [24]. However, these values may have been affected by the contribution of the synovium after the aspiration needle or may have been cut out during the download of the synovial fluid. This potential confounding factor was not controlled for. The lipid is present in the synovial fluid in the oligolamellar form [25] but it is not clear how these structures interact with the surface of the articular cartilage. We chose phosphatidylcholine for this study because the largest amount occurs in the synovial fluid.

The comparison of properties of natural and artificial articular cartilage surfaces in an aqueous two-phase lubricant is presented in Table 1 [2,3,16,23,26].

**Table 1.** Comparison of properties of natural and artificial articular cartilage surfaces in an aqueous two-phase lubricant.

| Properties | Natural Articular Cartilage | Artificial Articular Cartilage |
|---|---|---|
| Wettability | Surfaces are hydrophobic, $\Delta\theta \approx 0°$ | Surfaces with different hydrophobicity, $\Delta\theta = 22°–26°$ |
| Surface Property | Surface amorphous layer has a multi-bilayer structure and the surface is hydrophilic | A layer of protein adheres to the surface and this surface becomes hydrophilic |
| Friction | Ultra-low friction ($f = 0.005$) between tribopair with synovial fluid of 0.9% NaCl, lubricin and hyaluron, protein macromolecules, and phospholipid micelles, pH of ~7.4 | Low friction ($f = 0.05–0.15$) between tribopair with synovial fluid in the presence of 0.9% NaCl, lubricin and hyaluron, protein macromolecules, and phospholipid micelles, pH of ~7.2–7.4 |
| Degenerate Condition | Content of phospholipids is decreasing, phospholipids bilayers are uncertain or vanishing, wettability is lower (~40°) | Proteins more than phospholipids tend to adsorb strongly on soft and hard hydrophobic surfaces |

The model of interfacial tension compared to the pH of a spherical lipid bilayers membrane follows the graphs with a Gaussian function [18,27–29]. The effect of the pH of an electrolyte solution on the phosphatidylcholine bilayer lipid membrane was determined. The dependence of interfacial tension on the effect of pH of the electrolyte solution for phosphatidylcholine is presented with error bars in Figure 4. The experimental values are marked by points and the theoretical values are determined by a curve using Equation (2).

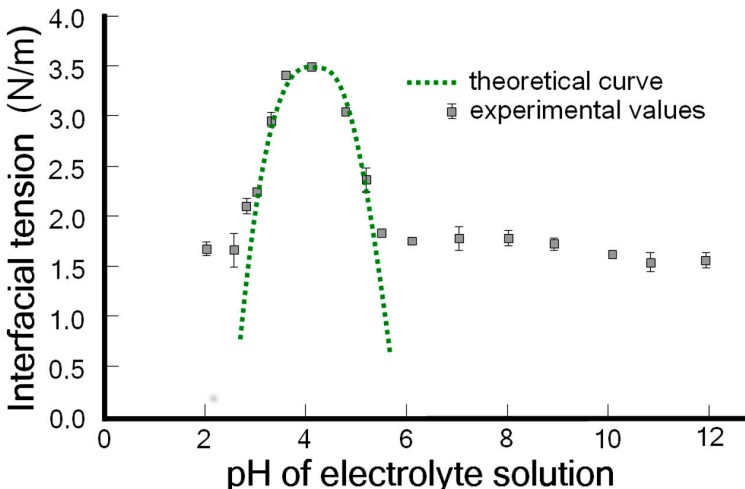

**Figure 4.** The dependence of the interfacial tension of a phosphatidylcholine bilayer versus pH of the electrolyte solution.

The curve demonstrates that the maximal interfacial tension values occur at the isoelectric point. The trends of these curves were well characterized by a simplified description based on the Gibbs isotherm but only in the proximity of the isoelectric point [18].

The interfacial tension value calculated from Equation (1) can substituted into Equation (2), where $K_a$ and $K_b$ values are determined using the method presented in reference [18].

$$\gamma = \gamma_{\max} + 2sRT \ln\left(\sqrt{\frac{K_a}{K_b}} + 1\right) - sRT \ln\left[\left(\frac{K_a}{a_{H^+}} + 1\right)\left(\frac{a_{H^+}}{K_b} + 1\right)\right] \tag{2}$$

where $\gamma$ [N/m] is the interfacial tension of the lipid membrane; $\gamma_{\max}$ is the maximal interfacial tension of the lipid membrane; $s$ [mol/m$^2$] is the surface PC concentration; $R$ is the gas constant; $T$ is the temperature; $K_a$ and $K_b$ are the acid and base association constants, respectively; and $a_{H^+}$ is the value of hydrogen ion concentrations.

At a solution of pH of ~1, the amino group of the phospholipids occurs in the protonated form [$(CH_3)_3N^+-$] and the interfacial tension is low. As the pH of the solution is increased, the amino group tends to lose some of its charge, i.e., $(CH_3)_3N^+- + -OH \rightarrow (CH_3)_3N^+OH^-$, leading to an increase in the interfacial tension towards a maximum value, which is namely the isoelectric point (IP). As the pH increases, the amino groups on the phosphatidylcholine surface gradually lose their positive charge and the ($-PO_4H$) groups lose protons ($-PO_4H \rightarrow -PO_4^-$), which leads to a negatively charged surface. The interfacial tension values are low.

Figure 5 shows the dependence of the friction coefficient for the mammal cartilage on time and pH. As shown in Figure 5, it can be observed that for different cartilages, there are different values observed for the friction. The curves in Figure 5 should be related to their amphoteric character and charge density. The presented results show that the friction coefficient is highly reliant on the electrostatic interaction between the two cartilage surfaces. The observed low friction of the two sliding cartilage sides that have the same charge is associated with the electrostatic repulsion of a charged cartilage/cartilage pair.

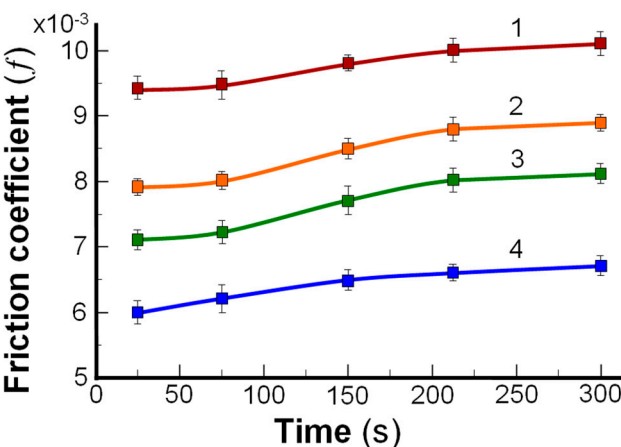

**Figure 5.** The friction coefficient versus time for the pair cartilage/cartilage in saline solution: 1, pH 4.2 (isoelectric point); 2, pH 6.0 (negatively charged surface); 3, pH 3.5 (positively charged surface); and 4, pH from 7.4 to 9.0 (negatively charged surface).

The changes in the articular cartilage charge density were caused by a variation in the pH of the electrolyte solution, which produced different charge densities of the articular cartilage. The curve was obtained at a pH of 3.5 before IP; for a pair of ($-NH_3^+/-NH_3^+$) charged positively with curves at the pH values of 6.0, 7.4, and 9.0 after IP; and for pairs of ($-PO_4^-/-PO_4^-$) charged negatively. Furthermore, the curve at a pH of 4.2 (IP) was obtained for no net charges.

It has been experimentally proven that the surface-active phospholipids present in the synovial fluid and on the surface of articular cartilage play an important role as a lubricant. When the total amount of phospholipids increased, the content of these phospholipids was reduced and the performance of the joints became poor [30,31].

In Figure 6, we show the self-organization of lipid vesicles in the synovial fluid, the formation of lamellar phases of phospholipids in the synovial fluid under a certain load, and the lamellar slippage of the phospholipid bilayers on the surface of cartilage under a certain load.

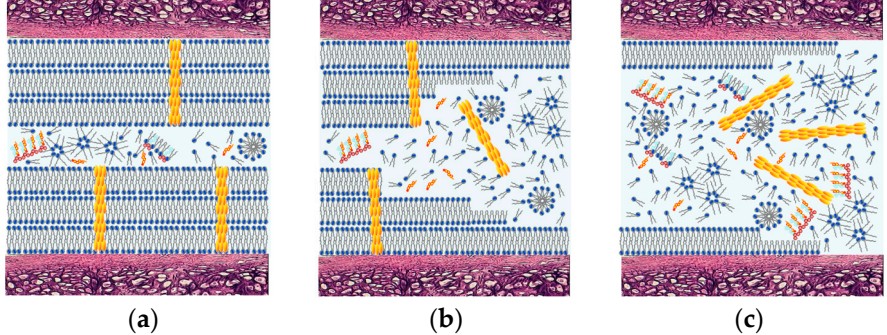

|  (a)  |  (b)  |  (c)  |

**Figure 6.** Phospholipidic (**a**) spheres and liposomes in synovial fluid; (**b**) lamellar phases under load in synovial fluid, and (**c**) phospholipid bilayers in lamellar slippage on the surface of cartilage under load.

The negatively charged surface of the cartilage with synovial fluid that has a pH of 7.4 is supported by lamellar slippage of bilayers when it is under a load. The measurements of the friction coefficient with liposomes, lamellar phases, and phospholipid bilayers [2] confirmed our hypothesis of low friction in the lamellar-repulsive mechanism [32].

This extremely low friction assigned to a low intermediate layer (contact angle ~0°), lamellar slippage of bilayers, and a short-range repulsion between the negatively charged ($-PO_4^-$) cartilage surfaces from the contribution of the hyaluronan and proteoglycans (PGs), which is a glycoprotein, are termed the lubricin and lamellar phospholipid phases [32].

This extremely low friction assigned to a low intermediate layer (contact angle ~0°), lamellar slippage of bilayers and a short-range repulsion between the negatively charged interfaces ($-PO_4^-$) cartilage surfaces from the contribution of the hyaluronan and proteoglycans (PGs), which is a glycoprotein, are termed the lubricin and lamellar phospholipid phases [32].

### 3.2. Hyaluronic Acid/Phosphatidylcholine Bilayer Interactions

Hyaluronic acid and phosphatidylcholine are important components of the lubricating molecular layer in joint synovial systems. Furthermore, due to their extensive profiles of biocompatibility and their recognized therapeutic applications, these molecules have been recognized as essential candidates for tribological surgical adjuvant design [33].

Phospholipids increase in the synovial fluid of osteoarthritis, resulting in a more significant number of forms aggregating, such as micelles and reverse micelles. This cannot provide a proper response to a specific external load because of the lack of a network of hyaluronic acid in preventing the collapse of micellar structures. Importantly, the mechanical damage to the continuity of the articular cartilage surface can be manifested by an increased pH of the synovial fluid and the concentration of the phospholipid molecule [34].

Hyaluronic acid is combined with an outer coating in the form of phosphatidylcholine vesicles in a solution of mass that can be deposited on the surfaces to easily deform other structures, such as multilayered structures, or wholly removed at higher loads. It was also found that when the phosphatidylcholine proceeds further into the solid phase, better lubrication results, although only a certain amount of phosphatidylcholine in the liquid disordered phase may be advantageous based on the expected load conditions [35,36].

The interactions between the phospholipids and compounds of hyaluronic acid fibers result in the cylindrical/micellar forms around these fibers which absorb the mass force applied to the articular cartilage, facilitating lubrication within the joint [37]. According to Pasquali-Ronchettiet et al. [38], the facilitated lubrication in this regime is obtained using a brush-like lubrication mechanism by which the phospholipid heads attach to hyaluronic acid and form inverted cylindrical micelles. This mechanism of hydration repellency may also be involved in the ability to increase the absorption of the force field when the hydrophilic heads repeat over each other to form a cushioning effect, which is supported by a suitable electrostatic shielded condition [39], in addition to the reactive force of the diaphragm discussed earlier [40]. Consequently, the reverse micelles can reduce the friction coefficient by changing the mode of friction, i.e., the quasi-static friction effect may be replaced by its rolling equivalent [41].

### 3.3. Lamellar-Repulsive Lubrication Mechanism between Cartilage/Cartilage Surface and Friction on Amphoteric Charged Surfaces

The joint's lubrication within the lamellar-repulsive mechanism is consistent with the literature data [3,42]. Some authors [3,42] have proposed that the lubrication is provided by phospholipids that form a multi-bilayer structure over the articular surface of cartilage; these phospholipids are known as surface active phospholipids. The lubrication mechanism in joints occurs both: (i) through lamellar lubrication, which occurs when the bilayers slide over each other, and (ii) through the structured synovial fluid when the lamellar spheres, liposomes, and macromolecules act like a roller-bearing between two cartilage surfaces to achieve effective biological lubrication [43]. The friction between the bovine cartilage/cartilage contact surfaces is highly sensitive to the pH of an aqueous solution in the pH range 1.0 to 6.5. The articular cartilage shows amphoteric properties with the gradual increase in friction ($-NH_3^+ \rightarrow -NH_2$), with a maximum at a pH of ~4.2 (the isoelectric point, Figure 4). After the IP, this leads to a negatively charged surface ($-PO_4H \rightarrow -PO_4^-$) with a decreased friction coefficient *f*.

The charge density of the cartilage surface was changed from positive to negative by varying the pH of the buffer solution. The positively ($-NH_3^+/-NH_3^+$) and negatively ($-PO_4^-/-PO_4^-$) charged tribological pairs showed lower values for the friction coefficient compared to the cartilage surface at

the isoelectric point (a maximum on the curve occurs under electric neutrality conditions) with a pH of ~4.2 (Figure 4). The low friction between the two cartilage surfaces demonstrates that the friction is mostly associated with their charge density by the electrostatic interaction between the two cartilage surfaces. The same charge of the cartilage surface should be attributed to the surface electrostatic repulsion of charges that are fixed on the AC. At a pH of 3.2, the positively charged ($-NH_3^+/-NH_3^+$) cartilage pair and the negatively charged ($-PO_4^-/-PO_4^-$) cartilage pair at a pH > 7 have comparable friction coefficients.

It can be observed that at a low charge density (log $[OH^-]$ = pCD) for the cartilage, the friction coefficient $f$ of the bovine cartilage versus the surface charge density decreases. Moreover, the friction decreases distinctly with an increase in the charge density of cartilage ($-OH^- + -PO_4H \rightarrow H_2O + -PO_4^-$).

At a higher charge density, the friction coefficient stabilizes, with the low slope possibly associated with the completed reaction of the phospholipids group ($-PO_4H \rightarrow -PO_4^-$), which indicates the absence of a protonated group ($-PO_4H$).

### 3.4. Deactivation of a Surface-Active Phospholipid Bilayer

The amphoteric nature of phospholipids allows them to self-assemble into a classic arrangement which represents a basic biological membrane. A surface-active phospholipid layer covers the natural articular surfaces in a multi-bilayer structure [44]. The well-defined outer layers of the bilayer are visible on excellent cartilage surfaces but osteoarthritis may result in the depletion of important joint molecules and surface-active phospholipids on the articular surface [2,3]. Furthermore, the depletion of the surface-active phospholipid lining was demonstrated by the cartilage wettability angle varying from 103° to 65°. This insight led to the hypothesis that the surface-active phospholipid is deactivated in the pathological state of osteoarthritis and remains present in the synovial fluid but in an inactive state (see Figure 7). The pathological synovial fluid contains three times more phospholipids but the cartilage structure changes and its ability to spread is fragile. During normal operation, the surface-active phospholipid serves as a two-layer perturbation victim and it can improve itself due to self-organization mechanisms.

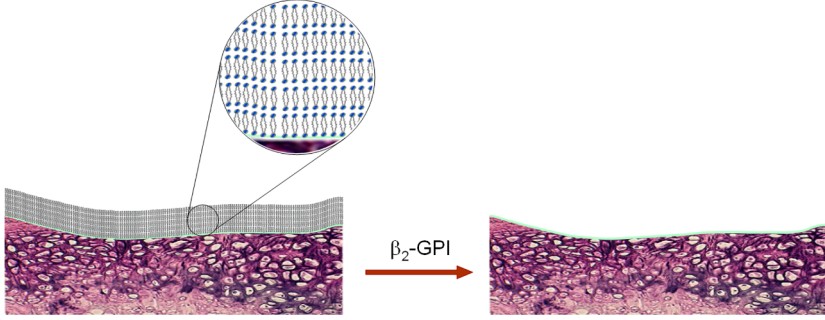

**Figure 7.** The open hockey stick-like conformation when β2-GP-1 is complexed to negatively charged phospholipids ($-PO_4^-$), which destroys the bilayers and deactivates phospholipid molecules.

The destruction of cartilage in most rheumatic diseases and osteoarthritis has generally been accepted as a mechanism for the inactivation of phospholipid bilayers. The acid–base interaction occurs between the protonated amino acid ($-NH_3^+$) group of β2-glycoprotein 1 and the phospholipid group ($-PO_4^-$), i.e., ($-NH_3^+$) + ($-PO_4^-$) $\rightarrow$ ($-NH_3 + PO_4^-$), which is strong enough to deactivate the surface of the phospholipid bilayer.

β2-Glycoprotein I (β2-GP-1) is a protein that circulates in the blood at various levels (50–500 μg mL$^{-1}$) with a molecular weight of 50 kDa. β2-Glycoprotein I can exist in a closed conformation, but when open, the confirmation resembles a hockey stick. β2-GP I in its hockey stick-like conformation is a highly adhesive protein and binds to different receptors on cells. The isoelectric point of β2-GPI occurs when the pH is 5–7, so the protein is negatively charged at a physiological pH of 7.4 [45]. This protein can interact

with the phospholipid membranes by electrostatic interactions and hydrophobic loops [44]. Binding β2-GP I to anionic charged phospholipid groups ($-PO_4^-$) at a pH of 7.4 causes a change in the protein conformation [46–48].

The softening of the cartilage is the first phase of cartilage deterioration [45,49]. Classical morphological changes of articular cartilage of bones and joints start with fibrillation and local disorganization of the surface, which results in the cleavage of cartilage surface layers. The new division occurs in the perpendicular direction to the surface of cartilage and is in line with the axes of "dominant collagen bundles." The continued deterioration of cartilage leads to subchondral bone exposure and more generalized synovial lesions. Understanding cartilage damage is essential for looking at the cellular processes and biochemical structure of healthy cartilage [49].

## 4. Conclusions

The changes in interfacial tension values are compatible with those in the phosphate- and amino-charged groups at a low pH (1.0 to 4.0) ($-NH_3^+ \rightarrow -NH_2$), after IP (pH 4.5), and at higher pH values (5.0 to 9.5) ($-PO_4H \rightarrow -PO_4^-$). The low friction is the result of the lamellar slippage of phospholipid bilayers and a short-range hydration repulsion between the negatively charged ($-PO_4^-$) cartilage surface and the proportion of macromolecules in the synovial fluid. The acid–base properties of the phospholipid bilayer significantly affect their wettability and surface friction properties.

The friction coefficient can change by as much as a factor of 10 for any phospholipid membrane. This leads to the lamellar repulsion mechanism under highly charged conditions but this quickly transforms into a stable low-friction state.

The open conformation of the hockey stick occurs when β2-GP-1 is complexed with negatively charged phospholipids ($-PO_4^-$), destroying bilayers and inactivating phospholipid molecules.

A tree of the 'lamellar-repulsive mechanism' of joint lubrication is based on a lamellar surface amorphous multilayer that is formed on the articular hydrophilic surface. The surface amorphous multilayer membrane provides lamellar-repulsive hydration lubrication on the boundary. The lamellar-repulsive mechanism is supported by phospholipid lamellar phases and charged macromolecules from synovial fluid between the charged cartilage surfaces.

**Author Contributions:** Conceptualization, A.D.P. and W.U.; Methodology, A.D.P.; Software, T.M.; Validation, K.J. and K.K.-D.; Formal Analysis, K.J. and K.K.-D.; Investigation, K.J. and K.K.-D.; Resources, T.M.; Data Curation, A.D.P. and W.U.; Writing—Original Draft Preparation, A.D.P. and W.U.; Writing—Review and Editing, A.D.P. and W.U.; Visualization, T.M.; Supervision, A.D.P. and W.U.; Project Administration, A.D.P. and W.U.; Funding Acquisition, W.U. All authors read, approved, and revised the final manuscript.

**Funding:** This research received no external funding.

**Conflicts of Interest:** The authors declare no conflict of interest.

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
