# Peer review of "Understanding the Unique Role of Phospholipids in the Lubrication of Natural Joints: An Interfacial Tension Study"

_coatings, doi:10.3390/coatings9040264_

Round 1
Reviewer 1 Report
In this paper, the authors report on the measurements on the lubricating aspects of an egg-based phospholipid molecule under different pH conditions. This is commonly found in joint lubrication. The work is interesting and may be suitable for the journal, Coatings but the authors are asked to make certain revisions before further consideration is given for publication in Coatings:
1. How does this lipid compare with Lubricin? Please discuss. Please compare the measured friction coefficient and hydrophobicity with other similar bio-based lubricants like lubricin.
2. The authors should include in their introduction and discussions a recent review article; Lubricants 2018, 6(2), 30; doi:10.3390/lubricants6020030 and clearly demonstrate where their contribution stands in view of all the works conducted so far on natural bio-lubricants.
3. Please tabulate measured friction coefficients.
4. Similarly, discussion lacks references to Hyaluronic acid, another important joint lubricant.
5. The authors should really introduce comparisons with respect to other famous joint lubricants while they discuss mechanisms associated with ‘lamellar-repulsive mechanism’!
6. Also, clarify better the effect of pH on friction coeff. I had a hard time trying to see this.
Author Response
Response to the Reviewer

Reviewer 2 Report
After a rapid and first reading of the manuscript entitled “Understanding the unique role of phospholipids in the lubrication of natural joints: the interfacial tension study”, I concluded that the present paper is very incomplete and it is written in a very carelessly way.
I think the paper is not in condition to be accepted for publication. I am sending in attachment a copy of the paper in which some comments/corrections are highlighted.
Kind regards,

Author Response
Response to the Reviewer

Reviewer 3 Report
The paper presents some important inaccuracies:
- In the Introduction paragraph (1.1) some phrases are probably reported from the instructions of the journal or from other documents: “The introduction should briefly place the study in a broad context and highlight why it is important. It should define the purpose of the work and its significance. The current state of the research field should be reviewed carefully and key publications cited. Please highlight controversial and diverging hypotheses when necessary. Finally, briefly mention the main aim of the work and highlight the principal conclusions. As far as possible, please keep the introduction comprehensible to scientists outside your particular field of research. References should be numbered in order of appearance and indicated by a numeral or numerals in square brackets, e.g., [1] or [2,3], or [4–6]. See the end of the document for further details on references. (lines 96-103);
- In the Conclusion paragraph, as final phrase of the paper: “This section is not mandatory, but can be added to the manuscript if the discussion is unusually long or complex”.
In Material and Methods, paragraph 2.1, the cited equation (1) is lacking.
The legend of Figure 4 is unclear: “The dependence of the interfacial tension of a lipid membrane made from PC on the pH of the electrolyte solution the experimental values are marked by points and the theoretical ones by curve”.
Similarly, several parts of the paper are not clear and need to be rewritten.
“The bilayers to be used for integration interfacial functions between surfaces and have been a subject of several inquiries due to its tribological features”. What does it mean? (lines 239-241).
And further: “However, in articular cartilage, the SAPL is absent because there is no suitable substrate on which the relevant lipid layer may be formed [16]. This statement contradicts what was stated above…. (in 216-218 lines).
“β2-Glycoprotein I (β2-GP I) can exist in closed conformation and open the confirmation resembles a hockey stick when β2-GP I in its hockey stick-like conformation is a highly adhesive protein and binds to different receptors on the cells Binding β2-GP I to anionic charged phospholipid groups (-PO4-) at pH ~ 7.4 causes the change of epitope conformation and exposure to autoantibodies [20-22]”. These sentences are very confuse…. (lines 260-264).
In general, the paper has to be reorganized: from the Result and Discussion section some parts should be moved in Material and Methods section, e.g. the entire 3.2. paragraph and the main part of the 3.3 paragraph.
Only a graph is reported in the Results and often are described qualitative aspect of the analyzed phenomenon. The most part of the comments in the “Results and Discussion” seem to be derived from the studies of other Autors; this would be right in a richer paper data, but in this case the obtained data are scares.
Author Response
Response to the Reviewer

Round 2
Reviewer 1 Report
It appears that the authors have made significant changes to their manuscript and improved it extensively taking into account all reviewer comments and concerns. The revised manuscript may now be acceptable for publication in the journal Coatings in its present form.
Reviewer 2 Report
In my opinion, the paper entitled “Understanding the unique role of phospholipids in the lubrication of natural joints: the interfacial tension study” is now ready to be accepted for publication.
Reviewer 3 Report
Please, check the numbering of "Results and Discussion" paragraph.
I appreciated the changes to the paper and I think that it is much improved.